# Regulatory T Cell Therapy of Graft-versus-Host Disease: Advances and Challenges

**DOI:** 10.3390/ijms22189676

**Published:** 2021-09-07

**Authors:** Mehrdad Hefazi, Sara Bolivar-Wagers, Bruce R. Blazar

**Affiliations:** 1Division of Hematology, Mayo Clinic, Rochester, MN 55905, USA; hefazitorghabeh.mehrdad@mayo.edu; 2Division of Blood and Marrow Transplant & Cellular Therapy, Department of Pediatrics, University of Minnesota, Minneapolis, MN 55454, USA; boliv004@umn.edu

**Keywords:** allogeneic hematopoietic stem cell transplantation, graft-versus-host disease, regulatory T cells, GVHD, Treg

## Abstract

Graft-versus-host disease (GVHD) is the leading cause of morbidity and mortality after allogeneic hematopoietic stem cell transplantation (allo-HSCT). Immunomodulation using regulatory T cells (Tregs) offers an exciting option to prevent and/or treat GVHD as these cells naturally function to maintain immune homeostasis, can induce tolerance following HSCT, and have a tissue reparative function. Studies to date have established a clinical safety profile for polyclonal Tregs. Functional enhancement through genetic engineering offers the possibility of improved potency, specificity, and persistence. In this review, we provide the most up to date preclinical and clinical data on Treg cell therapy with a particular focus on GVHD. We discuss the different Treg subtypes and highlight the pharmacological and genetic approaches under investigation to enhance the application of Tregs in allo-HSCT. Lastly, we discuss the remaining challenges for optimal clinical translation and provide insights as to future directions of the field.

## 1. Introduction

Allogeneic hematopoietic stem cell transplantation (allo-HSCT) is used as a curative therapy for various hematological and non-hematological disorders. However, its efficacy is limited by the occurrence of graft-versus-host disease (GVHD), which remains the leading cause of morbidity and mortality after allo-HSCT. Prophylactic GVHD regimens utilize nonspecific immune suppression, leaving cancer patients at risk of relapse, life-threatening infections, and drug toxicities. Corticosteroids remain the first-line treatment for both acute and chronic GVHD; however, less than 50% of patients achieve durable responses [1,2]. Currently, Ruxolitinib is the only U.S. Food and Drug Administration (FDA)-approved drug for acute GVHD in patients failing corticosteroids [3]. At this time, only Ibrutinib and Belumosudil (KD025) are FDA-approved drugs for chronic GVHD patients failing corticosteroids or >1 line of prior therapy [4,5]. Ruxolitinib is being evaluated by the FDA as consideration for approval based on the recently published REACH3 study (NCT03112603) [6]. New treatments are critically needed to reduce GVHD prevalence and severity in an effort to improve HSCT patient outcomes, as well as reduce toxicities associated with long-term drug therapy.

Regulatory T cells (Tregs) are critical for immune homeostasis and tolerance induction after HSCT. Anticipated advantages are permanency of effect, induction of tolerance that obviates the need for ongoing drug therapy, and potential for tissue repair [7,8,9]. Strategies to enhance Treg numbers and function have been adopted from preclinical studies and are showing potential in the clinic [10,11]. Preclinical studies have shown that the Treg function and specificity can be enhanced by pharmacological agents or via genetic modifications. This review highlights recent preclinical and clinical advances in Treg therapy for GVHD and discusses remaining challenges for clinical translation.

## 2. Treg Subtypes and Their Applications

CD4 Tregs are the most well-studied immune regulatory T cells, and are characterized by co-expression of CD4, CD25, and the transcription factor Forkhead box protein 3 (Foxp3) [12,13,14]. This subset in human comprise 2–4% of circulating CD4^+^ T cells; compared to 5–10% in secondary lymphoid organs of rodents [12,13,15,16,17]. Based on their origin, Tregs can be classified as those having developed in the thymus [known as thymic Tregs (tTregs) or natural Tregs (nTregs)] or induced in vivo in peripheral tissues from conventional T cells (Tcons) under non-inflammatory or inflammatory conditions and typically on mucosal surfaces [known as peripheral Tregs (pTregs)] or those induced from Tcons in vitro [known as induced Tregs (iTregs)] [18]. T cell receptors (TCRs) of tTregs generally recognize self-antigens, while iTregs tend to express TCRs specific for foreign antigens [19,20,21,22]. Tregs can be further classified through functional and phenotypic distinctions. The differentiation of Tregs is accompanied by functional diversification, which parallels the pattern of differentiation in Tcons [23,24,25,26,27]. These subsets of Treg cells specialize at the regulation of specific classes of immune responses. Different strategies are thus under investigation with the aim to tailor the in vitro polarization of Treg products for adoptive transfer in GVHD (Table 1).

The stability of Foxp3 expression is controlled by the methylation status of genes in a Treg-specific demethylation region (TSDR). The promotor and TSDR within the 2nd conserved non-coding sequence (CNS2) is completely demethylated in tTregs, resulting in a stable Foxp3 expression and suppressor function [28,29,30]. iTregs, on the other hand, lack the tTreg transcription factor signature (Eos, Lef1, Satb1, IRF4, GATA1; known as “locked-in” signature) and exhibit a partially demethylated TSDR [28,29,30,31]. As such, iTregs can be inherently unstable and lose their Foxp3 expression under certain proinflammatory conditions [32,33,34,35].

**Table 1 ijms-22-09676-t001:** Phenotypic and functional heterogeneity of regulatory T cells and their role in GVHD.

Subtype	Origin	Phenotype	Activation Factors	Suppressive Functions *	Role in HSCT
CD4 tTreg	Thymus	CD25^+^ Foxp3^+^ CD127^lo^ Helios^+^	IL-2 TCR/CD28	Cell-cell contact IL-10 TGF-β Poor Tcon priming by APCs	- Prevention of aGVHD and reversal of cGVHD when given as adoptive transfer [36,37,38,39]. - Either no effect or impairment of GVL [40,41]
CD8 iTreg	Periphery	CD25^+^ Foxp3^+^ CD127^lo^ Helios^−/+^	IL-2 TGFβ	Cell-cell contact IL-10 TGF-β	- Potential role in adoptive transfer for aGVHD prevention [42]
Type 1 Regulatory T Cell (Tr1)	Periphery	CD25^+/−^ Foxp3^−^ CD49β^+^ Lag3^+^	Tbet Blimp-1	IL-10 >> TGF-β Granzyme B CTLA-4	- Antigen-dependent suppression, hence reduced global immunosuppression [43,44] - Important role in aGVHD when donor tTregs are profoundly deficient [45] - Potential role in GVL [46,47]
Follicular Regulatory T Cell (TFR)	Thymus	CD25^lo^ Foxp^3+^ CXCR5^+^ Bcl6^+^ ICOS^hi^	mTOR STAT3 TCF-1	IL-10 Granzyme B TGF-β CTLA-4	- Important role in cGVHD and autoimmunity [48,49]
CD8 iTregs	Periphery	Foxp3^+^ CD28^−^	IL-2 TGFβ	IL-4 IL-10 TGF-β Perforin/Granzyme Fas-L/Fas	- Potential role in GVHD and GVL [45,50]

Abbreviations: aGVHD: acute GVHD; cGVHD: chronic GVHD; GVHD: graft-versus-host disease; GVL: graft-versus-leukemia; iTreg: induced Treg; tTreg: thymic-derived Treg. * In vitro and in vivo suppressor functions listed are not all inclusive.

### 2.1. CD4 tTregs

CD4 tTregs are the most well-studied immune regulatory T cells. It is clearly established that CD4 tTregs are essential for the maintenance of immune homeostasis and self-tolerance [51,52,53]. Mice lacking FoxP3 protein (scurfin), known as scurfy mice, have no Tregs, defective T cell tolerance and a generalized autoimmune disease that is severe and X-linked lymphoproliferative syndrome [12,13,16]. In patients FoxP3 mutation and Treg deficiency results in an autoimmune disease known as IPEX, immunodysregulation polyendocrinopathy enteropathy, X-linked [51].

Preclinical studies have shown that adoptive transfer of tTregs can control allograft rejection and GVHD by limiting alloimmune responses [54,55,56,57]. However, translation to the clinic has been challenging due to the low frequency of human tTregs (typically 2–4% of CD4^+^ T cells) in the peripheral blood, and that the phenotypic profile of human tTregs is not as readily distinguished in peripheral blood as in the spleen and lymph nodes of mice [15]. Currently, most studies isolate human Tregs from peripheral blood, although third-party umbilical cord blood (UCB) derived Tregs have been successfully used in clinical trials [36]. The advantages of using UCB are higher frequency of CD4^+^CD25^bright^ Treg cells and a reduced likelihood of isolating CD25^+^ Tcons due to the lower foreign antigen exposure and general immune regulatory environment of the fetus. In a phase I trial of ex vivo expanded polyclonal recipient Tregs infused into patients receiving living donor kidney transplant, the adoptive transfer of these CD4 iTregs post-transplant was safe with no adverse effects or rejection up to two years post-transplant [58].

Despite their relatively strong lineage fidelity, some but not all studies indicate that tTreg can lose stability and Foxp3 expression under certain conditions in an inflammatory environment, with consequent loss of immunosuppressive capacity [33,35,59,60]. Several groups have reported that TCR-stimulated tTregs can be converted to Th17 cells in the presence of IL-6 in vitro that can be reduced by exposure to TGFβ [61,62,63]. Decreased Foxp3 expression can cause immune disease by subverting the suppressive function of tTregs leading to Th2 cell generation [64]. Loss of function or elimination of Tregs expressing the Th1 associated T-box transcription factor T-bet results in severe Th1 autoimmunity [65]. Therefore, it is desirable to find an approach that can sustain the stability and function of tTregs in the inflammatory condition. Previous data have shown that all-trans retinoic acid (ATRA), the major vitamin A metabolite, increases histone acetylation on the Foxp3 gene promoter and suppresses differentiation of tTregs to Th17 cells in an inflammatory milieu [63,66,67]. Taken together, CD4 tTregs likely represent a more reliable regulatory cell type for adoptive transfer owing to their greater expansion potential, functional stability, and proven track record of safety in multiple clinical trials (see below).

### 2.2. CD4 iTregs

The relative scarcity of tTregs in the peripheral blood and the time and cost of in vitro Treg expansion protocols significantly limit their clinical usefulness. Thus, several groups have explored an alternative way to induce Tcons in vitro to generate iTregs. The main advantages of this approach are that iTregs can be generated in relatively high numbers, overcoming the limitations of the small starting population of tTregs and in vitro expansion and iTregs can be highly effective in tolerance induction to colitis that is further augmented by combining with tTregs [68,69]. CD4 iTregs can be generated under several different conditions, the most common of which include the culture of Tcons with TGF-β or retinoid acid. Foxp3 stabilization can also be bolstered by addition of compounds such as rapamycin, vitamin C that increases the activity of the ten-eleven translocation enzyme Tet2, or combination of demethylating agents and histone deacetylase inhibitors, amongst others [59,69,70,71,72,73,74,75,76]. Building on these efforts, Hippen and colleagues have developed a protocol for expanding large numbers of CD4 iTregs from Tcons in the presence of rapamycin, TGF-β, and IL-2, suitable for clinical trials [43].

### 2.3. CD8 iTregs

CD8 iTregs are induced in the periphery early after allo-HSCT and comprise a significant percentage of all Treg subsets [77]. The defining markers of CD8 iTregs are still under study and more research will be necessary to reach a consensus [78]. CD8 iTregs have been reported to engage in multiple cell-dependent and independent mechanisms to mediate immune suppression and homeostasis [79,80,81,82,83]. Using mice that were unable to generate both CD4 and CD8 iTregs, Beres et al. showed that iTreg-deficient mice had increased expansion of alloreactive T cells, and that CD8 iTregs in the absence of CD4 iTregs had the capacity to prevent GVHD severity [77]. Vitamin C given in vivo can induce and stabilize CD8 iTregs that diminish GVHD without loss of anti-tumor responses [84]. Deletion of Janus-activated kinase-2 (JAK2) in CD8 iTregs stabilized Foxp3 and increased their efficacy in preventing GVHD [84]. The adoptive transfer of in vitro generated CD8 iTregs can ameliorate GVHD by targeting alloreactive Tcon activation and proliferation through a CTLA-4 dependent mechanism [83]. The adoptive transfer of in vitro generated human CD8 iTregs have promoted tolerance without abrogating the graft anti-tumor responses in a humanized mouse model of GVHD [83]. Small molecule inhibitor of signal transducer and activator of transcription factor 3 (STAT3) polarized human iTregs generated from Tcons to express STAT5, augmenting iTreg stability and further reducing GVHD [85,86].

Antigen-specific CD4 iTregs selectively prevented GVHD via a mechanism of linked- suppression [50,87]. Adoptively transferred CD4 iTregs have a potent capacity to inhibit GVHD, but in the process, develop impaired graft-versus-leukemia (GVL) capacity [50]. In contrast CD8 iTregs decreased GVHD albeit more modestly than CD4 iTregs but maintained GVL responses [50]. Combined adoptive transfer of both CD4 and CD8 iTreg was superior to either cell type alone in improving the outcome of allo-HSCT by virtue of more potent GVHD and GVL effects [50]. Altogether, these findings highlight the role of iTregs in mitigating GVHD and offer potential strategies for its therapeutic use. Future work will involve the investigation of unique markers to identify iTreg subsets with their associated functions and identification of optimal strategies to induce, stabilize, and expand CD8 iTregs for use in the clinic.

### 2.4. Type 1 Regulatory (Tr1) Cells

Type 1 regulatory (Tr1) cells are a class of regulatory T cells, distinct from Tregs, with high IL-10, interferon-gamma (IFNγ), transforming growth factor-beta (TGFβ), no IL-4 or IL-17 and low/absent IL-2production; Tr1 cells express CD4^+^CD49β^+^Lag3^+^ and have transient and lower level FoxP3 with activation [88,89,90]. Tr1 cells require the transcription factors Tbet and B lymphocyte-induced maturation protein-1 (blimp-1),while the T-box transcription factor eomesodermin (Eomes) is essential for long term persistence [91]. Tr1 cells assist in maintaining immunological homeostasis and promoting tolerance [92]. The suppressive role of Tr1 cells is highlighted in settings with low to negligible levels of CD4^+^Foxp3^+^ Tregs. In acute GVHD, Tr1 become the main Treg subset and Tr1 deficiency leads to GVHD progression [91]. Tr1 cell can secrete TGFβ to inhibit both effector T cells and APCs, granzyme B to mediate APC cytotoxicity, and express cytotoxic T-lymphocyte associated protein 4 (CTLA-4) and programmed death 1 (PD-1) to target effector T cell activation [89,93,94].

To determine whether Tr1 cells could be used therapeutically, groups are using multiple strategies that range from in vitro gene editing to increasing Tr1 numbers in vivo. IL-10 and IL-10 inducing agents along with immunosuppressants, anti-CD3 antibodies, and tolerogenic DCs have shown promising results in Tr1 induction and GVHD prevention [89,90,95,96]. Tr1 cells are expressed at low frequency in vivo. Allogeneic donor CD4^+^ T cells cultured with host APCs in the presence of IL-10 generate Tr1 cells hyporesponsive to host alloantigens (termed Tallo-10 cells) [97,98]. IFNα is synergistic with IL-10 in inducing Tr1 cells from UCB [98]. Since IL-10 receptor expression is required for Tr1 cells, tolerogenic dendritic cells (DCs) that express immunoglobulin-like transcript 4 (ILT4), HLA-G and overexpress IL-10 have been used to drive the generation of Tr1 cells from Tcons. A sizable number of Tr1 can be generated in vitro from Tcons by inducing stable and sustained IL-10 overexpression in CD4 Tcons that suppress T cells in vivo while unexpectedly killing myeloid cells in an HLA class I-dependent but Ag-independent manner [47].

In preclinical model, Tr1 cells improved acute GVHD outcome [99]. In the clinic, delayed post-transplant in vivo infusion of Tallo-10 cells has been shown to be tolerogenic in some patients who had received haploidentical transplant [100]. The goal of current clinical trials is to determine whether Tr1 cell products are safe and tolerable. In T-allo10 trial (NCT03198234), a cell product containing 15% Tr1 cells is infused a day prior to patients undergoing allo-HSCT. Preliminary results are encouraging as the T-allo10 product is tolerable thus far and Tr1 cells appear to have long term persistence [101]. Future clinical trials will likely involve testing the safety and efficacy of genetically engineered Tr1 cells, antigen-specific Tr1 cells, and their applications beyond HSCT.

### 2.5. Follicular Regulatory T (TFR) Cells

Follicular regulatory T (TFR) cells share phenotypic characteristics with T follicular helper cells (TFH) and CD4 tTregs, yet TFR cells are very distinct from both. TFRs co-express Foxp3 and B-cell lymphoma protein-6 (Bcl6) and localize to germinal centers (GCs) due to their expression of CXCR5; however, they originate from thymic-derived Foxp3^+^ precursors, and not naïve or TFH cells [102]. Other cell surface determinants that are present on TFR cells are T cell inducible costimulator (ICOS), CD40L, and PD-1, with the latter restraining TFR function [103]. This specialized subset of regulatory cells prevents TFH cells from providing support to B cell maturation in GCs, thereby playing an important role in antibody production and regulation of autoimmunity. TFR cells suppress IL-21 and IL-4 production by TFH cells, and inhibit class switch recombination and antibody production by B cells [104,105,106].

The imbalance between TFH and TFR cells plays an important role in some models of chronic GVHD [48,49]. This is evidenced by an increase in TFH and B cells in GCs, along with low TFR cell and TFR/TFH ratio [48]. McDonald et al. showed that daily therapeutic IL-2 complexed with JES6- anti-IL2 antibody increased TFR frequency while TFH numbers were consequently reduced in mice [49]. Consistent with these data, chronic GVHD patients have high circulating TFH cells and are Treg deficient [107,108]. Low dose IL-2 infusion restored Treg homeostasis and diminished clinical manifestations [108,109]. Although ascribed to the preferential stimulation of CD25^hi^ TFR cells, TFRs do not express CD25 and hence do not have a high affinity IL-2 receptor. Sakaguchi and colleagues reported a distinct CD25^−^ TFR subpopulation [110]. At the same time, Klatzmann and co-workers showed that TFR cells are CD25^−^, while Botta et al. demonstrated that TFR cells are CD25 low and that IL-2 prevents TFRs from developing by Blimp-1 mechanism [111,112]. As the intermediate affinity IL-2 receptor beta chain (CD122) is expressed, IL-2 may support TFR expansion via an IL-2-STAT5 axis.

The immunomodulatory effect of TFR cells leads to a distinctive durable inhibition of TFH and B cells, while leaving global effector molecules and metabolic pathways intact [103]. TFRs may therefore be beneficial to patients with chronic GVHD, although obtaining sufficient number of TFR cells for adoptive transfer is an obvious challenge. With the advent of genetic engineering techniques, it may be possible to use polyclonal Tregs, and modify them so that they become TFRs. Indeed, Kim et al. showed that forced expression of CXRCR5 via retroviral transduction drives TFR-like features in Foxp3 Tregs. The expression of CXCR5 and Foxp3 appeared to be stable after adoptive transfer, leading to an effective suppression of antibody production from B cells stimulated with TFH cells [113]. Complementary or additive to low dose IL-2, the adoptive transfer of Tregs or the more specialized TFR cells could thus have a high potential in chronic GVHD, although more work is needed to determine how to best deploy TFR cells therapeutically.

## 3. Genetic Engineering Strategies

### 3.1. Antigen Specificity

Antigen-specific Tregs have garnered interest in the Treg field because of their increased potency compared to polyclonal Tregs on a per cell basis and their potential for reduced off-target immunosuppression [114]. Proof-of-principle experiments have shown that alloantigen-reactive Tregs can be expanded using donor APCs such as DCs or B cells [79,114,115,116]. Ex vivo expansion in the presence of the antigen of interest would accomplish this goal. Treg genome editing could be advantageous as it would confer antigen specificity to a larger Treg population rather than amplifying rare antigen-specific precursor cells achievable by introduction of engineered TCRs or a chimeric antigen receptor (CAR) that, for example, is directed to an MHC molecule [117,118,119].

#### 3.1.1. TCR-Engineered Tregs

The use of TCRs represents a physiological way of activating T cells and allows for targeting of intracellular antigens presented by HLA molecules. This approach has been successfully tested in various preclinical models of autoimmune disorders and transplantation [87,120,121,122,123]. Tsang et al. showed that adoptively transferred allo-specific murine CD4 tTregs generated by either donor APCs or transduction of a TCR reactive against donor antigens could promote indefinite heart allograft survival, even in completely mismatched mouse strains [123]. In another study chicken ovalbumin (OVA) specific CD4 iTregs generated from OVA-reactive OT-II TCR transgenic T cells efficiently prevented GVHD induced by polyclonal T effector cells in the allogeneic recipients that express OVA protein, but not in OVA(-) recipients [87]. The efficacy of these antigen-specific CD4 iTregs was significantly higher than polyclonal CD4 iTregs. While these results are encouraging, the main limitations of the TCR-engineering strategy are MHC restriction, which limits their coverage to a certain MHC-bearing host (allo-HSCT) or donor (solid organ graft), along with the risk of mispairing the engineered TCR with the endogenous TCR, which can cause undesired reactivity and off-target effects [124,125]. However, the expression of a single antigen-specific TCR in Tregs that is directed toward a donor-host antigen disparity may hinder GVHD prevention in situations in which tissue-specific antigens are not represented in culture systems or there is skewing to an in vitro immunodominant antigen that is subdominant in vivo and potentially diverting Tregs away from their target antigens.

#### 3.1.2. Chimeric Antigen Receptor (CAR) Tregs

The success of CAR T cells in hematological malignancies has generated interest in redirecting Treg specificity to inciting antigens that contribute to autoimmunity and transplantation rejection [126,127]. CARs are synthetic receptors that consist of an extracellular single chain variable fragment (scFv) linked with an intracellular CD3 activation domain. Depending on the CAR generation one or more costimulatory domains will deliver intracellular signals to T cells. CARs are advantageous compared to TCR- guided approaches because of their ability to recognize antigen directly on the cell surface in a non-MHC restricted manner [128]. Recently, groups have targeted the human leukocyte antigen (HLA) present on allografts and absent in recipients to redirect Treg specificity to promote graft tolerance and reduce GVHD [118,119,129,130]. For example, anti-HLA-A2 CAR Tregs demonstrated superior function compared to polyclonal Tregs in suppressing xenogeneic GVHD and more potently reduced skin allograft rejection compared to polyclonal Tregs [118,119,129,130]. A phase 1/2a trial examining the safety profile of autologous anti-HLA-A2 CAR Treg cells for patients undergoing kidney transplantation is ongoing (NCT04817774).

Another antigen recently tested to redirect CAR Tregs in preclinical studies is the CD19 antigen expressed on B cells. Anti-CD19 CAR Tregs have been applied in humanized mouse GVHD models to demonstrate potent suppression of xenogeneic GVHD clinical manifestations (weights; GVHD scores), as well as demonstrated efficacy to decrease skin allograft rejection [131]. Ongoing efforts are focused on evaluating optimal CAR design for Tregs as different costimulatory domains affect Treg phenotype, function, and cytokine production [132,133]. Boroughs et al. found that the use of a 4-1BB-based CAR in Tregs reduced expression of suppressive cytokines and negatively affected their function in vitro and in vivo [132]. Confirming and expanding on these studies, Dawson et al. performed an extensive study on how different costimulatory domains, including CD28, ICOS, CTLA4, PD-1, GITR, OX40, 4-1BB, and TNFR2, modulate the function of an anti- HLA-A2 CAR in an allotransplantation model [133]. In contrast to anti-tumor CAR studies, the data showed that the CD28-encoding CAR was superior in vitro and in vivo in terms of proliferation, suppression and delay of GVHD symptoms. Interestingly, the 4-1BB-CAR and TNFR2-CAR expression negatively affected Treg function and stability, inducing methylation of the Foxp3 locus, downregulating the expression of Helios, and reducing suppressive function in vitro and in vivo [133]. In contrast to the findings of Dawson et al., Koristka et al. used a modular CAR technology called UniCAR and showed that CD28-based CARs might exert off-target activity and heightened cytolytic activity compared to 4-1BB based CARs [134]. CAR Treg design will continue to be optimized for potency, specificity, and persistence.

An important concern with adoptive Treg cell therapy is that infused cell product may contain potentially deleterious Tcon cells. Even when Tcon cells are rigorously excluded from the infusion product that can be accomplished by high-speed sorting as used by Orca Bio (final Orca-T product ~93.8% Tregs), the safety of adoptively transferred Tregs may be challenged by the propensity of some Tregs to differentiate into Tcons under an inflammatory condition chrome [135]. The risk is arguably higher when Tregs are genetically edited to redirect their specificity. In fact, lineage tracing experiments have shown that Tregs carrying a TCR specific for an antigen expressed in the central nervous system had converted to Tcons and precipitated paralysis in a mouse model of multiple sclerosis [124]. Similar results were reported in a mouse model for autoimmune arthritis [136]. Therefore, production of genetically modified antigen-specific Tregs will require the use of improved protocols for the purification and amplification of Tregs to prevent contaminating Tcons with putative hazard.

### 3.2. Foxp3 Gene Editing of Tcons to Generate Tregs

Given the key role of Foxp3 in controlling Treg function, several groups have used genetic engineering to increase or stabilize its expression [137,138,139,140]. This strategy could allow high numbers of Tregs to be generated from Tcons, hence circumventing the need for isolation and expansion of polyclonal Tregs. Functional CD4^+^ Tregs from IPEX patients can be generated by ectopic expression of Foxp3 in Tcons [140]. Further, CRISPR-based Foxp3 gene editing by precise homology directed repair (HDR) was used to obtain Foxp3 expression in hematopoietic stem and progenitor cells (HSPCs) from IPEX patients; in vitro regulatory function of the edited cells was observed and HSPCs differentiation potential was preserved [138]. Edited cells demonstrated stable Foxp3 expression under inflammatory conditions and suppressed xenograft acute GVHD [141]. In a separate study, Wright and colleagues showed that retroviral mediated gene co-transfer of Foxp3 and a TCR could convert CD4^+^ Tcons into antigen-specific Tregs in mice capable of suppressing immune responses by T cells specific for a third-party antigen, albeit less potent than TCR-transduced natural Tregs [142]. The difference between engineered natural Tregs and Foxp3-converted T cells transduced with the same TCR may lie in the inability of the latter to exploit endogenous promoter and enhancer regions or a requirement for other factors to stabilize the Treg phenotype. To test this hypothesis, Fu et al. used a computational and experimental approach to reverse engineer the transcriptional regulatory network of Tregs [143]. These experiments showed that Foxp3 alone was not sufficient to “lock in” Treg cell signature, but a partially induced Treg profile could be stabilized by co-transduction with one of five transcription factors Eos, IRF4, GATA-1, Lef1, or Satb1 [143].

Epigenetic editing technology has also been used to upregulate or stabilize Foxp3 expression. Delivery of catalytically inactive CRISPR-Cas9 (dCas9) fused to the catalytic domain of histone acetyltransferase showed that histone acetylation targeted to the promoter locus was able to activate and stabilize Foxp3 levels in mice, even under inflammatory conditions [144]. Similarly, Foxp3 expression could be upregulated via dCas9 fused to a transcriptional activator and guide RNAs recognizing the Foxp3 promoter [145]. In contrast, Kressler et al. showed that epigenetic editing via CRISPR-dCas9-TET1 leads to stable TSDR demethylation and Foxp3 expression, but is not sufficient to convert Tcons into a fully functional Treg [146]. Such epigenetic editing approaches may thus need to be combined with other methods that augment iTreg function or stability.

### 3.3. iPSC-Derived Tregs

Induced pluripotent stem cells (iPSC) can be used as an alternative source of Tregs. Human iPSCs are generated from fibroblasts or cord blood cells and transduced with defined transcription factors to revert them into pluripotent stem cells [147]. These cells can then be used to form almost all cells of the body. In mice, iPCS-derived Tregs can be generated through transduction of Foxp3 and co-culture with stromal cells that express Notch ligand [148,149]. Additionally, TCR and Foxp3 gene-transduced iPSCs can be used to differentiate antigen-specific iPSC-Tregs. Adoptive transfer of antigen-specific iPSC derived Tregs has proven effective in a well-established antigen-induced arthritis model, as well as in a murine model of autoimmune diabetes [148,149]. As protocols for generating human iPSC-derived Tregs are being developed, comparative studies of CD4 tTregs and CD8 iTregs in terms of potency, stability, and longevity will be critical.

### 3.4. Drug Resistant Tregs

After allo-HSCT, many patients require ongoing immunosuppression to prevent or treat GVHD. Immunosuppressive drugs such as Cyclosporin, Tacrolimus, and corticosteroids can potently suppress T cell function and provoke lymphocyte apoptosis [150,151]. To address this limitation, several groups have genetically engineered T cells for resistant to these drugs, thereby permitting their preferential survival. Large-scale GMP-compliant CRISP/Cas9-mediated knock out of the glucocorticoid receptor and tacrolimus-binding protein has been used in the context of virus specific T cells to confer drug resistance [152,153,154]. Similar strategies may be applied to adoptive Treg therapy in the future.

## 4. IL-2 Modulation Strategies

Exogenous IL-2 administration has been ascribed to be deleterious in GVHD owing to its ability to promote effector T cell function, although when given in high doses for several days beginning on transplant day 0, allo-HSCT recipient mice had markedly diminished GVHD [155,156]. As a consequence of high-affinity CD25 expression on Tregs and NK cells compared to Tcons, low or ultra-low dose IL-2 preferentially stimulated Treg and CD56^hi^CD16^−^ NK cell proliferation and promoted Treg stability without augmenting cytotoxic T cells [108,109,157,158,159,160]. Clinical trials have shown that prophylactic low dose IL-2 administration during the early post-transplant period could effectively enhance Treg expansion and decrease the incidence of acute and chronic GVHD [158,161]. An open-label controlled randomized trial of low-dose (1 × 10^6^ U/d) given for 2 weeks beginning on day 60 in allo-HSCT recipients showed that the IL-2 treated group had a significantly lower incidence of moderate-to-severe chronic GVHD (33% vs. 57%), accompanied by a significant increase in GVHD-free and GVHD progression-free survival at three years (47% vs. 31%) [161]. Low-dose IL-2 therapy also has been shown to be efficacious in the treatment of steroid-refractory chronic GVHD in phase I/II clinical trials [108,109,162].

IL-2 may activate effector T cells that have the potential to worsen GVHD and NK cells which have a low GVHD capacity [163]. IL-2 has a relatively short half-life (<30 min) and is rapidly excreted in the urine. Tregs can be preferentially stimulated by administering IL-2/anti-IL-2 antibody complexes or IL-2 fusion protein that has a prolonged half-life and lower IL-2 peak levels [164,165,166]. This approach has shown efficacy in experimental models of autoimmune diseases and solid organ transplantation [166,167,168,169]. In the context of HSCT, preclinical data has shown that IL-2/anti-IL-2 antibody complexes have the capacity to expand Tregs and ameliorate chronic GVHD, a low inflammatory disease, while aggravating acute GVHD, a high inflammatory disease [49]. Studies on the binding interaction of IL-2 with anti-IL-2 mAbs have shown that IL-2 can be mutated to selectively stimulate Tregs, CD8 T cells, or NK cells [160]. Selective Treg stimulation can be favored by mutating IL-2 to bind poorly to IL-2R-β while retaining IL-2 binding to IL-2R-α chain [170]. Ongoing phase I/II clinical trials are currently testing molecularly engineered IL-2 with an increased half-life in patients with steroid-refractory chronic GVHD (NCT03422627) or systemic lupus erythematosus (NCT04680637 and NCT04433585).

In a different approach, investigators engineered a mutated IL-2 receptor beta chain (IL-2Rβ, CD122) that binds its IL-2 ortholog with high affinity. Transduction of the mutated IL-2R into tTregs enabled selective stimulation by exposure to orthogonal IL-2 in vitro and in vivo, with limited off-target effects and toxicity due to the negligible binding to a wildtype IL-2 receptor, suggesting a clinical strategy [171]. In a similar study, Hirai et al. introduced an orthogonal IL-2R-β chain into Tregs and demonstrated that upon adoptive transfer in a murine mixed hematopoietic chimerism model, orthogonal IL-2 injection significantly promoted orthogonal IL-2R^+^ Treg proliferation without increasing other T cell subsets [172]. Under nonmyeloablative conditions, this strategy facilitated donor hematopoietic cell engraftment followed by the acceptance of heart allografts.

## 5. Treg Immunometabolism and GVHD

Recent attention has been placed on understanding the immunometabolism of Tcons and Tregs, including those present during GVHD. Specific metabolic perturbations have been reported during GVHD [173,174,175,176,177]. While the metabolic profiles of effector T cells (Tcon) in GVHD are becoming clearer, much is left to understand regarding Treg generation, proliferation, stability, and suppression. T cells increase aerobic glycolysis, mitochondrial oxidative phosphorylation (OXPHOS), and glutaminolysis early after allo-HSCT [173,174,175,176,177,178,179]. Unlike Tcons, Tregs preferentially use OXPHOS to exert their suppressive functions [180,181]. However, Tregs also need glycolysis to fuel the biosynthetic and energetic needs associated with proliferation and homing [182,183].

There is an interest in targeting metabolic pathways that preferentially modulate Treg while reducing or at least not potentiating alloreactive Tcons. One commonly used approach is mTOR inhibition, which targets glycolysis in rapidly proliferating alloreactive T cells during GVHD [184]. Rapamycin, an mTOR inhibitor, has been particularly successful compared to other immunosuppressants because of its potential to inhibit Tcons and to enhance Foxp3 expression and Treg suppressive function [185,186]. Treg access to, deficiency of and excess exposure to specific substrates and metabolites can alter Treg phenotype, function, proliferation, homing, and survival. The host microbiome production of unique metabolites can be protective during HSCT by shaping pTreg generation. Treg function can be potentiated in vivo by enhancing fatty acid oxidation (FAO) that can fuel OXPHOS. Butyrate, a short chain FA produced by *Clostridia*, increased colonic Tregs, although butyrate can also be acute GVHD protective in a Treg independent manner [187,188,189,190,191]. In chronic GVHD, high circulating levels of microbe-derived short chain FAs are associated with GVHD protection. Metformin, a 5’ AMP-activated protein kinase (AMPK) activator and FAO promoting agent, led to decreased Th17 cells, increased Tregs, and reduced acute GVHD severity and prevalence [192]. Inhibiting fatty acid synthesis significantly decreased Th17 effectors while supporting the development of Foxp3^+^ Tregs. Additionally, inhibiting glutamine transporters or glutamine or glutamate metabolism shifted from Th1/Th17 effector T cells towards increased Treg [193,194,195].

Foxp3 expression suppresses glycolysis and drives OXPHOS. Murine iTreg induction relies on lipid oxidation for OXPHOS. In contrast to iTregs, murine tTregs engage in glycolysis and glutaminolysis at comparable levels to Teff cells despite maintained Foxp3 expression [196]. Exposure of murine tTregs to TGF-β repressed phosphotidylinositol 3-kinase (PI3K)-mediated mTOR signaling, inhibited glucose transporter and the first enzymatic step in glycolysis, hexokinase-2 expression, resulting in metabolic reprogramming to favor OXPHOS [196]. In human ex vivo cultures, tTregs and iTregs utilize glycolysis upon activation. Nonetheless, inhibiting glucose metabolism by exposure of human tTregs and iTregs to 2-deoxy-D-glucose (2DG) had distinct effects [197]. At the time of human tTreg activation, 2DG treatment mediated by mTORC1 significantly reduced the proliferation and suppressor molecule expression (ICOS, CTLA-4) with minimal FOXP3 expression [197]. In contrast, 2DG modestly decreased the proliferation of iTregs during the induction phase and strongly reduced ICOS and FOXP3 expression. The addition of 2DG on day 3 post activation did not impact proliferation for either tTregs and iTregs. In contrast, adding 2DG to Th0 cultures impaired their expansion but not cytokine production. These data provide strategies to improve Treg generation and expansion, while minimizing naïve T cell expansion, pointing to an approach that would inhibit contaminating Teff cells but spare human tTregs and iTregs.

Treg metabolism also can be indirectly altered by targeting intracellular pathways and cellular structures. For example, localization of protein kinase C-theta (PKC-θ) in Toncs is at the immunological synapse, the point of contact between T cells and APCs where bidirectional signaling can occur. In Tregs, PKC-θ is located in a distal pole complex. Disruption of the PKC-θ distal pole complex by a small molecule inhibitor such as AEB071 increased in vitro and in vivo Treg suppressor function associated with reduced mTORC2 and increased metabolic fitness, as evidenced by increased expression of nutrient receptors (and FA uptake) and the rate-limiting enzyme for FA oxidation, and OXPHOS [38]. Similar results were seen using an siRNA to vimentin, a cytoskeletal intermediate filament that can tether mitochondria, inhibiting mitochondria network formation by the fusion of membranes of two distinct mitochondria into a single energy efficient mitochondria.

In summary, immunometabolism offers a new avenue to potentially prevent and treat GVHD, while providing ways to enhance Treg cell therapies. A particular advantage of this approach is that many metabolic modulators are already approved by the U.S. FDA or in clinical trials and could be repurposed to expedite new therapies into clinic to improve HSCT patient outcomes.

## 6. Translational Challenges

The manufacturing process for adoptive cell therapy is known to be complex and costly, which subsequently influences the translational success of these products. Important challenges particular to the field of Treg therapy are discussed below.

### 6.1. Treg Isolation and Purity

A major Treg challenge is obtaining high-level purification from an infrequent starting population using good manufacturing process (GMP) techniques. Treg isolation can be done using magnetic (MACS) or fluorescence-activated cell sorting (FACS) [10,42,198]. MACS is performed in a closed, sterile system, which involves clinical-scale magnetic enrichment of cells and can quickly process high numbers of untouched cells. Although magnetic bead sorting for CD4^+^/CD25^hi^ cells may incorporate CD127^lo^, purity may remain inadequate because activated naive and especially memory Tcons express CD25 and are therefore difficult to separate by MACS from CD25^hi^ Treg population. Including more surface markers (e.g., CD45RA^+^, CD49d^−^, and CD39^hi^), can result in a better defined and purer Treg population; however, when using MACS, there is a limit to the number of markers that can be included [199,200,201,202].

Compared to MACS, enrichment of Tregs by flow cytometry sorting enhances post-sort purity by allowing for additional selection markers and employing antigen density-defined cut-offs. This however significantly adds to the logistics and cost of Treg isolation including droplet-based sorting under GMP conditions and the requirement for GMP antibodies and flow sorter [203,204]. Currently, FACS equipment is not accepted as GMP-grade by many European regulatory authorities and is not suitable for scaling up in late-stage clinical trials. The newest generation of such droplet-based cell sorters attempts to fulfill GMP requirements, and these are being used in two clinical trials for allo-HSCT (NCT03802695 and NCT04013685) [205]. Another potential alternative could be the use of microfluidic switch technologies, such as the MACSQuant^®®^ Tyto^®®^ cell sorting platform, and fluid-channel sorters and technology developed by Orca Bio [37,135].

### 6.2. Ex Vivo Expansion and Stability

Freshly sorted Tregs have been used in the clinic [206]. Previous studies have demonstrated that a high ratio of Tregs to Tcons (e.g., 1:1 or 1:2) may be needed to mediate tolerance [36,42,206,207,208]. In lymphopenic allo-HSCT recipients, the transfer of freshly sorted Tregs two days before Tcons allowes for in vivo expansion. Myeloablation induces the highest degree of lymphopenia. For patients with autoimmunity, solid organ transplant or non-myeloablated allo-HSCT, it is uncertain that the degree of Treg expansion will be sufficient to control disease. Alternatively, Tregs can be ex vivo expanded using IL-2 and either anti-CD3/28 antibody-coated beads or anti-CD3 antibody loaded, K562 cell-based artificial antigen-presenting cells APCs (aAPCs) that express the high affinity Fc receptor CD64 and the costimulatory molecule CD86 (termed KT64/86) [43]. As compared to anti-CD3/28 antibody beads, cell-based aAPCs can significantly increase tTreg yield, while dominantly maintaining Foxp3 expression and suppressive function [43,209]. Total peripheral blood tTreg expansion induced by successive stimulations with KT64/86 cells was 200-fold higher compared with anti-CD3/28 beads (25,000 fold vs. 5 million-fold, respectively) [43].

Prolonged ex vivo expansion with repeated stimulations can have a negative impact on Treg Foxp3 expression and suppressive function [101,210,211]. Adding rapamycin to the culture can mitigate these effects, albeit at the cost of cell yield [43]. Peripheral blood tTregs stimulated two times without rapamycin (resulting in ~10 000-fold expansion) or five times with rapamycin (resulting in ~10,000,000-fold expansion) continued to preferentially express the defined Treg “locked in” factors. Importantly, Tregs cultured in rapamycin did not increase exhaustion gene expression even after five stimulations, in contrast to Tregs stimulated two times without rapamycin [212]. To enhance Treg stability during ex vivo expansion, Miyara and colleagues used a combined drug regimen consisting of IL-2, rapamycin, a pan-histone deacetylase inhibitor (vorinostat), and a hyomethylating epigenetic modifier (5-azacitidine). The combination drastically increased Foxp3 stability with an expansion fold that was only xx? times less than in the presence of IL-2 alone with an overall expansion of 37-fold, considerably lower than other published protocols [213].

With this degree of Treg expansion and since third-party Tregs can effectively suppress acute GVHD in mice, an “off-the-shelf” Treg bank stocked with cryopreserved Treg aliquots is feasible [214]. With current methodologies, Tregs temporarily lose suppressor function upon thawing. While complicating the distribution of Tregs for infusion, function can be restored after overnight IL-2 exposure, supporting the creation of a readily available cell bank [43]. Others have shown that thawing of cryopreserved expanded Tregs followed by restimulation can overcome the detrimental effects of cryopreservation on Treg number and phenotype [215]. In other studies, higher Treg viability and Foxp3 expression were obtained when cells were cryopreserved 1–3 days, but not >3 days after last restimulation [216]. The ability to cryopreserve and thaw expanded Tregs will have broad-ranging implications when designing clinical trials.

### 6.3. Concurrent Immunosuppressive Drugs

In allo-HSCT patients, Tregs may rapidly disappear from the peripheral blood, often ascribed to the lack of IL-2 released by T effectors that have been suppressed or use of immunosuppressive drugs, such as calcineurin inhibitors (CNI), that inhibit IL-2 production [10,217]. Unlike CNIs, rapamycin preferentially supports Treg expansion due to the differential sensitivity of Teffs vs. Tregs to mTOR inhibition [218,219]. As discussed above, rapamycin has been shown to stabilize the suppressor function and gene expression profile of Tregs, both for endogenous and adoptively-transferred Tregs [220]. Therefore, rapamycin is typically favored over CNIs for clinical trials when adoptive Treg therapy is added to standard-of-care GVHD prophylaxis. Furthermore, nonhuman primate studies have shown that, rapamycin could promote long-term persistence of adoptively transferred Tregs when combined with IL-2 or OX40L blockade [221]. Other pharmacologic agents such as post-transplant cyclophosphamide, histone deacetylase inhibitors (e.g., vorinostat), hypomethylating agents (e.g., azacitidine), JAK1/2 inhibitors (e.g., Ruxolitinib), and ROCK1/2 inhibitors (e.g., Belumosudil) have also demonstrated the ability to increase Treg compartment after allo-HSCT [221,222,223,224,225,226,227,228]. Thus, these agents are at least theoretically less likely to negatively impact the efficacy of Treg therapy when given as an adjunct immunomodulatory agent for GVHD.

### 6.4. Clinical Efficacy and Adverse Effects

Initial GVHD therapy studies by Tzonkowski and subsequent phase I/Ib GVHD prevention trials have confirmed the feasibility and safety of adoptive Treg therapy in GVHD (Table 2) [10,11,36,42,198,199]. The first acute GVHD prevention clinical studies were reported by two groups [36,206]. In our study, UCB was used as the source of tTregs in a phase 1 dose-escalation study [36]. Ex vivo expansion with anti-CD3/anti-CD28 antibody-coated beads achieved the targeted Treg dose in 74% of cultures with retention of suppressive function in all products. Twenty-three patients received a dose of 0.1–30 × 10^5^ UCB Treg/kg after double UCB transplantation. Adoptive transfer of Tregs in patients receiving mycophenolate and sirolimus/ cyclosporin reduced the incidence of grade II-IV GVHD compared to historical controls (43% vs. 61%) [36]. In a second study by our group, tTreg expansion was significantly increased via restimulation with aAPC, resulting in a median expansion of 13,000-fold. In the context of mycophenolate and sirolimus immunosuppression, adoptive transfer of Tregs virtually eliminated grade II-IV acute GVHD, with a cumulative incidence of 9% at 100 days [10].

In a study by Di Ianni and coworkers, tTregs were freshly isolated from peripheral blood and allowed to become activated and expanded in vivo prior to the infusion of haploidentical T cells. Despite lack of post-transplant immunosuppression, only two of 26 evaluable patients developed ≥ grade 2 acute GVHD, and no patient developed chronic GVHD after a median follow-up of 11.2 months [206]. In a more recent study by the same group, 43 patients with high-risk acute leukemia underwent haplo-identical HSCT without post-transplant immunosuppression and received manipulated grafts, containing CD34+ cells (mean 9.7 × 10^6^/kg), Tregs (mean 2.5 × 10^6^/kg), and Tcons (mean 1.1 × 106/kg). Even though 1.1 × 10^6^/kg Tcons were infused without any in vivo GVHD prophylaxis, the incidence of grade 2 acute GVHD was 15%, which was similar to the 11% in historical controls. More importantly, the cumulative incidence of relapse was only 5%, which was significantly lower than in historical controls [11].

The clinical efficacy of adoptive Treg therapy has been less encouraging when used for the treatment rather than the prevention of acute GVHD. The first-in-man clinical trial using ex vivo expanded tTregs for the treatment of GVHD demonstrated significant alleviation of the symptoms in one patient with chronic GVHD, but only transiently improved acute GVHD in a second patient [198]. In another study by Theil et al., adoptive transfer of ex vivo expanded tTregs resulted in clinical response in two out of five patients with refractory chronic GVHD [229].

Large-scale iTreg expansion methods facilitated a first-in-human phase 1 trial of CD4 iTregs that has been completed [43]. A cell dose of 3 × 10^8^/kg iTregs (7:1 ratio with Tcons) was safely given as GVHD prophylaxis to HSCT recipients of HLA-matched sibling donors along with cyclosporin and mycophenolate with a reduction, albeit non-significant, in acute GVHD incidence [42]. In tTreg and iTreg trials, neither relapse rates nor the incidence of opportunistic infections have been observed to be increased in small human studies reported to date. Overall, these early phase clinical trials demonstrate that adoptive Treg therapy is a feasible and safe therapeutic approach and may be efficacious especially for GVHD prevention. Larger randomized trials are needed to verify these findings and establish efficacy.

**Table 2 ijms-22-09676-t002:** Completed and ongoing clinical trials involving adoptive Treg therapy in GVHD (ClinicalTrials.gov; search date 30 July 2021).

Study ID	Ph	Enrollment Actual/Planned	HSCT/IST	Indication	Cell Product/Dose	Outcomes	Status	Center	References
NCT01634217	I	16/16	MRD MMF/Siro, *n* = 2 MMF/CSA, *n* = 14	GVHD PPx	Expanded PB CD4 iTregs Dose-escalation 3 × 10^6^/kg–3 × 10^8^/kg	Final Results: 1st cohort: 100% grade 3 aGVHD 2nd cohort: 20% ≥ grade 2 aGVHD	Completed	University of Minnesota, USA	[42]
NCT02423915	I	5/5	dUCBT, *n* = 2 PB MUD, *n* = 3 MMF/Siro	GVHD PPx	Fucosylated fresh UCB CD4 tTreg 1 × 10^6^/kg–1 × 10^7^/kg	Final Results: 100% ≥ grade 2 aGVHD	Completed	MD Anderson, USA	[230]
NCT01660607	I/II	12/24	TCD MRD/MUD No IST, *n* = 5 TAC or Siro, *n* = 7 Remaining, *n* = 12	GVHD PPx	Fresh CD4 tTregs and Tcons Dose-escalation and extension 1 × 10^6^/kg–3 × 10^6^/kg Treg and 1 × 10^5^/kg–3 × 10^7^/kg Tcon	Interim Results: 1st cohort: 40% ≥ grade 2 aGVHD 2nd cohort: No GVHD (n = 7)	Recruiting	Stanford, USA	[39]
NCT01795573	I	38/48	MRD IST: unknown	GVHD PPx	Donor CD4 tTregs expanded with recipient DCs Dose: unknown	No results	Active, not recruiting	Moffitt Cancer Center, USA	N/A
2012-002685-12	I	9/9	Not specified	GVHD PPx	Fresh CD4 tTreg Up to 5 × 10^6^/kg × once	Final Results: Safe; not designed for efficacy	Completed	University Hospital Regensburg, Germany	[231]
01/08	I	28/28	Haploidentical Without IST	GVHD PPx	Fresh PB CD4 tTregs and Tcons Dose-escalation 2 × 10^6^/kg–4 × 10^6^/kg Treg and 0.5 × 10^6^/kg–2 × 10^6^/kg Tcon	Final Results: 15% developed ≥ grade 2 aGVHD 5% developed relapse	Completed	University of Perugia, Italy	[11,206]
NCT00602693	I	11/11	dUCBT MMF/Siro	GVHD PPx	Expanded UCB CD4 tTreg Dose-escalation 3 × 10^6^–1 × 10^8^/kg Treg	Final Results: 9% developed ≥ grade 2 aGVHD 5% developed relapse	Completed	University of Minnesota, USA	[10]
NCT00602693	I	23/23	dUCBT	GVHD PPx	Expanded UCB CD4 tTreg Dose-escalation 0.1 × 10^5^–30 × 10^5^/kg Treg	Final Results: 43% ≥ grade 2 aGVHD (vs. 61% in historical control)	Completed	University of Minnesota, USA	[36]
NCT04678401	I	NA/10	Haplo Without IST	GVHD PPx	Treg enriched haplo graft NA: Treg dose	No results	Recruiting	Dana-Farber Cancer Institute, USA	N/A
NCT04013685	I	NA/84	TCD MRD/MUD Single agent IST	GVHD PPx	Engineered donor graft: TCD graft with additional infusion of Tcon and Tregs (TregGraft/Orca-T)	No results	Recruiting	Multi-center, USA	N/A
NCT01903473	II	NA/35 Treg arm, *n* = 10 Control, *n* = 25	Any	SR cGVHD	Fresh PB CD4 tTregs with sirolimus and low-dose IL-2 0.5 × 10^6^/kg × once	No results	Recruiting	University of Liege, Belgium	N/A
NCT01937468	I	NA/25	Any	SR cGVHD	Fresh PB CD4 tTreg Dose: Unknown	No results	Active, not recruiting	Dana-Farber Cancer Institute, USA	N/A
NCT02385019	I/II	NA/22	Any	SR cGVHD	Fresh PB CD4 tTreg Dose-escalation and extension 0.5 × 10^6^/kg–3 × 10^6^/kg × once	No results	Unknown	IMMJLA, Portugal	N/A
NCT01911039	I	NA/20	Any	SR cGVHD	Unknown Treg Type Dose-escalation 1 × 10^5^/kg–1.5 × 10^6^/kg × once	No results	Unknown	Stanford University, USA	NA
NCT02749084	I/II	NA/20	Any	SR cGVHD	Multiple donor PB CD4 tTreg Dose-escalation 1.7 × 10^5^/kg–6.6 × 10^6^/kg monthly × 3	No results	Recruiting	Universitaria di Bologna, Italy	NA
EK 206082008	I	5/5	Any	SR cGVHD	Expanded PB CD4 tTreg Dose-escalation 5 × 10^5^/kg–4.4 × 10^6^/kg × once	Final Results: Clinical response in 2 pts Stable disease in 3 pts	Completed	University Hospital Carl Gustav Carus, Germany	[229]
NCT03683498	I	0/16	Any	SR cGVHD	Expanded CD4 tTreg Dose-escalation 0.5 × 10^6^/kg–2 × 10^6^/kg	No Results	Active, not recruiting	FPAGI, Spain	N/A
NCT01911039	I	NA/20	Any	SR cGVHD	Unknown type Dose-escalation 1 × 10^5^/kg–1.5 × 10^6^/kg	No Results	Unknown	Stanford University, USA	N/A
NKEBN/458-310/2008	I	2/2	MRD Single agent IST	SR aGVHD SR cGVHD	Expanded CD4 tTreg 1 × 10^5^/kg in SR cGVHD 3 × 10^6^/kg in SR aGVHD	Final Results: Reduced IST in cGVHD Only transient improvement in aGVHD	Completed	Medical University of Gdańsk, Poland	[198]
NCT01453140	I/II	3/15	Any	SR aGVHD	Expanded PB CD4 tTreg with cyclophosphamide and sirolimus with or without Azacitidine Unknown dose	No results	Completed	John Theurer Cancer Center, USA	N/A

Abbreviations: aGVHD: acute GVHD; cGVHD: chronic GVHD; CSA: cyclosporin; dUCBT: double umbilical cord blood transplant; GVHD: graft-versus-host disease; HSCT: hematopoietic stem cell transplantation; IST: immunosuppressive therapy; MMF: mycophenolate mofetil; MRD: matched related donor; MUD: matched unrelated donor; NA: not available; PB: peripheral blood; Siro: sirolimus; SR GVHD: steroid-refractory GVHD; TAC: tacrolimus; UCB: umbilical cord blood.

## 7. Conclusions and Future Directions

The past decade has seen substantial advances in our understanding of the Tregs biology and mechanisms of tolerance induction after allo-HSCT. Preclinical and clinical research have established the potential of Treg based therapies as a promising alternative to pharmacologic immunosuppression for GVHD. Although early phase clinical trials have confirmed the feasibility and tolerability of adoptive Treg therapy, few data are available regarding the efficacy and reproducibility of this approach in late stage randomized clinical trials. Treg cell therapy for GVHD has not yet entered clinical practice due to several remaining obstacles including the high dose of polyclonal Tregs required to prevent/treat GVHD, difficulty of ex vivo Treg expansion, and absence of phase III clinical data. It is likely that genetic engineering of Tregs with an enhanced immunosuppressive capacity and an improved ability to expand and exert targeted function will enable their use as a promising therapeutic modality for GVHD. However, more questions remain open regarding potency, specificity, and functional differences of engineered Tregs, as well as the optimal gene delivery techniques (viral or non-viral systems) and manufacturing platforms.

Several start-up companies have recently been funded with the aim of applying genetically engineered Tregs to treat autoimmune and alloimmune conditions [232]. The first CAR-Treg clinical trial has been granted authorization in the UK and the Netherlands (STEADFAST) applying anti-HLA-A2 CD4 Tregs in kidney transplant patients (EUCTR2019-001730-34-NL). The use of gene editing as a tool for generating off-the-shelf CAR-T cells is very promising and can be translated to CAR-Treg therapy. This can be achieved by using CRISPR/Cas9 or transcription-activator-like effector nucleases (TALENs) to knock out the TCRα chain (TRAC) or β2 microglobulin of the MHC molecule, to prevent alloreactive T cells from inducing GvHD [233,234]. Most CAR Treg studies to date have used the so-called “second-generation” CAR design, which is used in the context of oncology to deliver potent Tcons for tumor eradication. Tregs and Tcons, however, have distinct requirements for suppression/effector function, and thus a future direction is to further optimize CAR design for optimal suppression as opposed to cytotoxicity. Furthermore, Tregs can be modified using new engineering strategies, such as Notch receptors that have an extracellular single-chain antibody and intracellular transcriptional domains that are released and activate expression of target genes [235]. Novel delivery methods such as nanoparticle-based approaches may facilitate precise targeting of cell types, such as Tregs that need to be enhanced in order to restore a tolerizing milieu in target tissues. It should also be noted that other regulatory cell types exist and have shown promise as potential therapeutic tools, including tolerogenic dendritic cells, natural killer cells, regulatory B cells and myeloid-derived suppressive cells [45,236]. Human and/or patient organoids, may gain more importance and are promising candidates for examining Treg function in disease models [237]. Furthermore, biomarker studies will be important to define not only the effects of Treg therapy but also the timing and doses of their administration. Overall, the development and production of a successful Treg therapy continues to represent an exciting and challenging endeavor, and one that offers hope for future therapies to be more targeted and efficacious.

## Data Availability

Not applicable.

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
