# Peer review of "Regulatory T Cell Therapy of Graft-versus-Host Disease: Advances and Challenges"

_ijms, 2021, doi:10.3390/ijms22189676_

Round 1
Reviewer 1 Report
Title: Regulatory T Cell Therapy of Graft-versus-Host Disease: Advances and Challenges
From the literature, we are well aware that cell therapy with polyclonal regulatory T cells (Tregs) has been translated into the clinic and is currently being tested in transplant recipients and patients suffering from autoimmune diseases. Moreover, building on animal models, it has been widely reported that antigen-specific Tregs are functionally superior to polyclonal Tregs. Among various options to confer target specificity to Tregs, genetic engineering is a particularly timely one as has been demonstrated in the treatment of hematological malignancies where it is in routine clinical use. Genetic engineering can be exploited to express chimeric antigen receptors (CAR) in Tregs, and this has been successfully demonstrated to be robust in preclinical studies across various animal disease models. However, there are several caveats and a number of strategies should be considered to further improve on targeting, efficacy and to understand the in vivo distribution and fate of CAR-Tregs
In this review, the authors provide the most up to date preclinical and clinical data on Treg cell therapy with a particular focus on GVHD. The authors also discuss the different Treg subtypes and highlight pharmacological and genetic approaches under investigation to enhance the application of Tregs in allo-HSCT. The topics presented are:
- Treg Subtypes and Their Applications
2.1. CD4 tTregs
2.2. CD4 iTregs
2.3. CD8 iTregs
2.4. Type 1 Regulatory (Tr1) Cells
2.5. Follicular Regulatory T (TFR) Cells
- Genetic Engineering Strategies
3.1. Antigen Specificity
3.1.1. TCR-Engineered Tregs
3.1.2. Chimeric Antigen Receptor (CAR) Tregs
3.2. FoxP3 gene editing of Tcons to generate Tregs
3.3. iPSC-Derived Tregs
3.4. Drug Resistant Tregs
- IL-2 Modulation Strategies
- Treg Immunometabolism and GVHD 421
- Translational Challenges
6.1. Treg Isolation and Purity
6.2. Ex Vivo Expansion and Stability
6.3. Concurrent Immunosuppressive Drugs
6.4. Clinical Efficacy and Adverse Effects
- Conclusion
I have some comments/suggestions which the authors can consider:
One of the stated objectives was to discuss remaining challenges for optimal clinical translation and provide insights as to future directions of the field.
Nevertheless, I feel this part is not well addressed
I was wondering what is the new knowledge/message that this review offers in comparison with these recent reviews:
The Future of Regulatory T Cell Therapy: Promises and Challenges of Implementing CAR Technology
https://www.frontiersin.org/articles/10.3389/fimmu.2020.01608/full
Regulatory T-Cell Therapy for Graft-versus-host Disease
https://www.ncbi.nlm.nih.gov/pmc/articles/PMC5049884/
Feasibility, long‐term safety, and immune monitoring of regulatory T cell therapy in living donor kidney transplant recipients
https://onlinelibrary.wiley.com/doi/abs/10.1111/ajt.16395
Figure 1. Phenotypic and functional heterogeneity of regulatory T cells and their role in GVHD.
The figures of the subtype seems redundant and similar among all subtypes/
Table 1. Completed and ongoing clinical trials involving adoptive Treg therapy in GVHD (search date July 30, 2021).
Comments: Did you mention which database used for the search?
Foxp3 expression suppresses glycolysis and drives OXPHOS. iTreg induction relies on lipid oxidation for OXPHOS. In contrast to iTregs, murine tTregs engage in glycolysis and glutaminolysis at comparable levels to Teff cells despite maintained Foxp3 expression [187]. –pls check citation.
Author Response
I have some comments/suggestions which the authors can consider:
One of the stated objectives was to discuss remaining challenges for optimal clinical translation and provide insights as to future directions of the field.
Nevertheless, I feel this part is not well addressed
We thank the reviewer for the comment. We have now added a full paragraph under section 7 (Conclusions and future directions) to discuss the future directions of the field. This is in addition to the discussion of ongoing challenges under each specific section.
I was wondering what is the new knowledge/message that this review offers in comparison with these recent reviews:
The Future of Regulatory T Cell Therapy: Promises and Challenges of Implementing CAR Technology
https://www.frontiersin.org/articles/10.3389/fimmu.2020.01608/full
Regulatory T-Cell Therapy for Graft-versus-host Disease
https://www.ncbi.nlm.nih.gov/pmc/articles/PMC5049884/
Feasibility, long‐term safety, and immune monitoring of regulatory T cell therapy in living donor kidney transplant recipients
https://onlinelibrary.wiley.com/doi/abs/10.1111/ajt.16395
We thank the reviewer for the comment. The above-mentioned review by Mohseni et al. focuses only on CAR-Tregs, whereas in our review we are covering several other aspects of Treg therapy for GVHD, including different subtypes of polyclonal Tregs, TCR-engineered Tregs, Foxp3-engineered Tregs, iPSC-derived Tregs, IL-2 modulation strategies and metabolic reprogramming of Tregs. Under the CAR-Treg section specifically, we have described more recent findings from three publications that are not included in the review by Mohseni et al.:
- Dawson et al’s study of eight different costimulatory domains in HLA-A2 CAR-Tregs (Ref 124).
- Interim results of the safety, feasibility, and clinical efficacy of a Treg-engineered donor product, called Orca-T (Ref 126).
- The study by Imura et al. showing potent suppression of xenogenic GVHD and skin allograft rejection by anti-CD19-CAR Tregs (Ref 122).
The review by Heinrichs et al. is from 2016. In our review, we have discussed more recent studies, including over 90 publications since 2016. The review by Harden et al. focuses on Treg therapy in Kidney Transplantation, whereas our review is primarily focused on GVHD. The inflammatory environment differs qualitatively (reagents that successfully lock kidney graft rejection are not uniformly extrapolatable to GVHD) and quantitatively (intense in GVHD; localized in kidney transplants), and alloantigen that drives immune responses would be localized (kidney) or broadly distributed (GVHD).
Figure 1. Phenotypic and functional heterogeneity of regulatory T cells and their role in GVHD.
The figures of the subtype seems redundant and similar among all subtypes/
We thank the reviewer for the suggestion. We have now modified Figure 1 into a table format (Table 1).
Table 1. Completed and ongoing clinical trials involving adoptive Treg therapy in GVHD (search date July 30, 2021).
Comments: Did you mention which database used for the search?
We thank the reviewer for the comment. The database used for the search is now added to the table.
Foxp3 expression suppresses glycolysis and drives OXPHOS. iTreg induction relies on lipid oxidation for OXPHOS. In contrast to iTregs, murine tTregs engage in glycolysis and glutaminolysis at comparable levels to Teff cells despite maintained Foxp3 expression [187]. –pls check citation.
We thank the reviewer for comment. Citation [187] is correct and corresponds with PMID 30209190. doi:10.4049/jimmunol.1800311 for murine Tregs. For human Tregs, we also cite reference 188 that demonstrates species-specific differences in glycolysis dependency.
Reviewer 2 Report
I consider your work very valuable, well structured and important for everyone involved in the field of transplantation.
Author Response
I consider your work very valuable, well structured and important for everyone involved in the field of transplantation.
We appreciate the reviewer’s feedback.
Round 2
Reviewer 1 Report
The edited version has improved the overall quality of the manuscript. I have no further comment.